🔓 | Open Peer Review | Antimicrobial Chemotherapy | Research Article

# Central role of the *ramAR* locus in the multidrug resistance in ESBL-*Enterobacterales*

François Gravey,[1] Alice Michel,[2] Bénédicte Langlois,[1] Mattéo Gérard,[2] Sébastien Galopin,[2] Clément Gakuba,[3] Damien Du Cheyron,[4] Laura Fazilleau,[5] David Brossier,[6] François Guérin,[7] Jean-Christophe Giard,[2] Simon Le Hello[1]

**ABSTRACT**    The aim of this study was to evaluate the proportion of resistance to a temocillin, tigecycline, ciprofloxacin, and chloramphenicol phenotype called t2c2 that resulted from mutations within the *ramAR* locus among extended-spectrum β-lactamases-*Enterobacterales* (ESBL-E) isolated in three intensive care units for 3 years in a French university hospital. Two parallel approaches were performed on all 443 ESBL-E included: (i) the minimal inhibitory concentrations of temocillin, tigecycline, ciprofloxacin, and chloramphenicol were determined and (ii) the genomes obtained from the Illumina sequencing platform were analyzed to determine multilocus sequence types, resistomes, and diversity of several *tetR*-associated genes including *ramAR* operon. Among the 443 ESBL-E strains included, isolates of *Escherichia coli* (*n* = 194), *Klebsiella pneumoniae* (*n* = 122), and *Enterobacter cloacae* complex (*Ecc*) (*n* = 127) were found. Thirty-one ESBL-E strains (7%), 16 *K. pneumoniae* (13.1%), and 15 *Ecc* (11.8%) presented the t2c2 phenotype in addition to their ESBL profile, whereas no *E. coli* presented these resistances. The t2c2 phenotype was invariably reversible by the addition of Phe-Arg-β-naphthylamide, indicating a role of resistance-nodulation-division pumps in these observations. Mutations associated with the t2c2 phenotype were restricted to RamR, the *ramAR* intergenic region (IR), and AcrR. Mutations in RamR consisted of C- or N-terminal deletions and amino acid substitutions inside its DNA-binding domain or within key sites of protein–substrate interactions. The *ramAR* IR showed nucleotide substitutions involved in the RamR DNA-binding domain. This diversity of sequences suggested that RamR and the *ramAR* IR represent major genetic events for bacterial antimicrobial resistance.

**IMPORTANCE**    Morbimortality caused by infectious diseases is very high among patients hospitalized in intensive care units (ICUs). A part of these outcomes can be explained by antibiotic resistance, which delays the appropriate therapy. The transferable antibiotic resistance gene is a well-known mechanism to explain the high rate of multidrug resistance (MDR) bacteria in ICUs. This study describes the prevalence of chromosomal mutations, which led to additional antibiotic resistance among MDR bacteria. More than 12% of *Klebsiella pneumoniae* and *Enterobacter cloacae* complex strains presented mutations within the *ramAR* locus associated with a dysregulation of an efflux pump called AcrAB-TolC and a porin: OmpF. These dysregulations led to an increase in antibiotic output notably tigecycline, ciprofloxacin, and chloramphenicol associated with a decrease of input for beta-lactam, especially temocillin. Mutations within transcriptional regulators such as *ramAR* locus played a major role in antibiotic resistance dissemination and need to be further explored.

**KEYWORDS**    gram-negative bacteria, enterobacteriaceae, genomics, mechanisms of resistance, extended-spectrum beta-lactamase, RamR, RamA, efflux pumps, intensive care units, regulation pathway, clinical microbiology, AcrAB-TolC

Address correspondence to François Gravey, gravey-f@chu-caen.fr.

François Gravey and Alice Michel contributed equally to this article. Author order was determined both alphabetically and in order of seniority.

The authors declare no conflict of interest.

*E*nterobacterales are Gram-negative bacteria that tend to develop resistance to many antibiotics commonly used in clinical practices, including last-resort molecules such as carbapenems, colistin, and aminoglycosides, leading to a major public health concern (1, 2). Antimicrobial resistance mechanisms through transferable resistance genes are well known, but chromosomal mutations are also involved in several ways: (i) by changing the target of the antibiotics or (ii) by interfering with cellular pathways such as efflux pump regulation and expression (3, 4).

Among *Enterobacterales*, resistance-nodulation-division (RND) pumps have been associated with antimicrobial resistance phenotypes, strain fitness, and virulence, especially AcrAB-TolC (5, 6). Regulation of this tripartite efflux pump is complex and mainly involves three major transcriptional regulator systems. AcrR is the local repressor that controls the transcription of *acrAB* (6). The expression of *acrR* is itself regulated by two physically distant operon products MarRAB and RamRA, which also control the expression of *tolC* (6).

*ramAR* locus, which does not exist in *Escherichia coli*, has been widely described among several species like *Salmonella enterica* serovar Typhimurium (7), *Enterobacter cloacae* complex (*Ecc*) (4), *Klebsiella pneumoniae* (8), and *Klebsiella aerogenes* (9) whereas *marRAB* has been well studied in *E. coli* (10). AcrR, RamR, and MaR belong to the TetR/AcrR family. Transcriptional regulators are organized in dimers, and their secondary structures are exclusively composed of alpha helices (11). They are divided into two functional parts: (i) the N-terminus forms a helix turn helix (HTH) pattern, which is a DNA-binding domain (12), and (ii) the C-terminus of the protein constitutes the protein–substrate interaction interface (12). Both RamR and MarR act as constitutive inhibitors of the expression of *ramA* and *marA* (6), which products belong to the AraC/XylS family (11). RamA and MarA behave as positive transcriptional regulators of many genes, including *acrAB* and *tolC*.

Moreover, the *acrAB* transcript can be protected by carbon storage regulator A (CsrA). CsrA is an RNA-binding protein; it is interacting with the 5′ extremity of the *acrAB* transcript, which stabilizes its secondary structure. The *acrA*, *acrB,* and *tolC* messenger RNAs are then protected from the formation of a repressive structure, allowing a correct interaction with the ribosome and maximizing the expression of efflux pump-encoding genes (13). In addition, other regulatory systems, such as SoxS and Rob, can also recognize the same DNA sequence as RamA and play a role in the transcription of *acrAB-tolC* (6).

Mutation within the N-terminal part of RamR among a clinical extended-spectrum beta-lactamase (ESBL) producer *Enterobacter hormaechei* strain has been recently described and led to additional resistance to antibiotics, including four classes of antibiotics, temocillin (TEM), tigecycline (TIG), ciprofloxacin (CIP), and chloramphenicol (CHL) called the t2c2 phenotype in the present study. These four resistances were related to *acrAB-tolC* overexpression and *ompF* downregulation due to RamR mutation (4).

The impact of antimicrobial therapy by cefoxitin, quinolones, and fluoroquinolones on selecting RND-overexpressing bacteria has already been described (14, 15). Patients hospitalized in intensive care units (ICUs) are usually under important selective pressure with the intensive use of antimicrobial and nonantimicrobial molecules; these factors can lead to the selection of strains that overexpress their RND pumps.

The aim of this study was to identify the proportion of clinical ESBL-*Enterobacterales* (ESBL-E) that showed a t2c2 phenotype among a collection of 443 strains isolated from unique patients in three ICUs in a French university hospital for 3 years that could be explained by mutations within the *ramAR* locus and/or other *tetR*-associated genes.

## RESULTS

### Diversity of strains and β-lactamase descriptions

A total of 476 ESBL-E were isolated in the three ICUs between 2019 and 2021. Ninety-three percent (*n* = 443) belong to *E. coli*, *K. pneumoniae,* or *E. cloacae* complex (*Ecc*) species and were included in this study. Strains came from the surgical ICU (*n* = 245,

55%), medical ICU ($n = 178$, 40%), and neonatology unit ($n = 20$, 4%) (Table S1). Eighty-two percent of the strains ($n = 364$) were isolated from rectal swab samples, 5% from urine samples ($n = 24$), and 3% from blood cultures ($n = 15$) (Table S1). The species distribution was *E. coli* ($n = 194$, 44%), *K. pneumoniae* ($n = 122$, 28%), and *Ecc* ($n = 127$, 28%). There was an important diversity within the species; indeed, 69 unique sequence types (STs) were found among the *E. coli* strains and 27 STs and 16 STs for *K. pneumoniae* and *Ecc*, respectively (Table S1). Regardless of the species, the major ESBL encoding genes found were *blaCTX-M-15* ($n = 307$, 70%), followed by *blaCTX-M-27* ($n = 34$, 8%) and *blaSHV-12* ($n = 30$, 7%). These data show a great diversity of bacterial populations and resistomes among the strains analyzed.

## Minimum inhibitory concentration determination and impact of Phe-Arg-β-naphthylamide

The minimum inhibitory concentrations (MICs) of the 443 strains are available in Table S1. A total of 35 strains ($n = 19$ for *K. pneumoniae* and $n = 16$ for *E. hormaechei*) showed resistances to CIP, TEM, and TIG according to the results obtained with the Sensititre system. Interestingly, no strain of *E. coli* was simultaneously resistant to the three molecules. Determined by the broth microdilution method, the MICs of CHL revealed that 15 out of 16 *E. hormaechei* and 16 out of 19 *K. pneumoniae* strains were also resistant to this antibiotic (Table 1). As a consequence, 12% (15/127) of *Ecc* strains and 13% (16/122) of *K. pneumoniae* strains presented the t2c2 phenotype.

Resistant strains were not associated with a specific ICU and/or a specific sample origin (data not shown).

MICs of TEM varied between 16 and 128 mg/L, those for TIG were between 2 and 4 mg/L, while those for CIP were from 0.5 to 256 mg/L, and those for CHL varied between 16 and >256 mg/L (Table 1). Elevated MICs observed for CIP and CHL can be explained by the acquisition of additional resistance mechanisms such as *aac(6')-Ib-cr*, *qnr* genes, and/or mutations within the quinolone resistance determining region (QRDR) for CIP and the presence of *cat* and *floR* genes for CHL (Table 1).

The addition of 20 mg/L of PAβN to the medium significantly decreased the MICs of CHL, CIP, and TIG (Table 1). CHL showed the greatest MIC reduction, between 3 and 4 dilutions, followed by CIP (between 0 and 4 dilutions) and TIG (between 1 and 3 dilutions) (Table 1). These results strongly suggested the involvement of an RND pump in the phenotypes observed. As mentioned in Materials and Methods, the MICs of TEM were not determined in the presence of PAβN, as the resistance of this molecule has previously been associated with the downregulation of *ompF*, which is not modified by the addition of PAβN.

## Sequence extraction and diversity of RND-associated regulators

The 443 isolates were sequenced and analyzed. The sequences of 12 genes linked to the expression of RND pumps (*ramA*, *ramR*, *acrR*, *acrA*, *acrB*, *tolC*, *soxR*, *soxS*, *marA*, *marB*, *marR*, and *csrA*) and the *ramAR* intergenic region (IR) were found among the 122 *K. pneumoniae* and 127 *Ecc* strains. As expected, the *ramAR* operon was absent from the 194 genomes of *E. coli* studied. The sequences of the proteins and the *ramAR* IR showed tremendous diversity, which was more important among *Ecc* strains ($n = 124$ unique sequences) than in *K. pneumoniae* strains ($n = 70$ unique sequences) (Fig. 1; Tables S2 and S3). In both species, sequences of RamR and the *ramAR* IR have been the most variable elements. Indeed, 24 and 23 unique sequences of the *ramAR* IR and 17 and 12 unique sequences of RamR were found among the *Ecc* and *K. pneumoniae* strains, respectively (Fig. 1; Tables S2 and S3). In contrast, the proteins CsrA, MarA, SoxS, and RamA were the most conserved elements among the two species. These results highlighted that within a regulatory pathway, some elements presented more important sequence diversity. These variable genetic elements could be a factor in bacterial adaptation and phenotype modification in ICUs.

**TABLE 1** Origins, sequence types, resistomes, and manual minimum inhibitory concentrations with or without 20 mg/L of PAβN for temocillin, ciprofloxacin, tigecycline, and chloramphenicol in the 31 resistant strains[b]

| Species | Study name | ST | Origin | Acquired beta-lactamase | Temocillin[a] | Fluoroquinolones AG + QRDR mutations[c] | Ciprofloxacin[a] (+PAβN) | Tigecycline AG[c] | Tigecycline[a] (+PAβN) | Chloramphenicol AG[c] | Chloramphenicol[a] (+PAβN) |
|---|---|---|---|---|---|---|---|---|---|---|---|
| *Enterobacter cloacae* complex | Ecc2020803 | ST78 | Pus | blaSHV-12, blaTEM-1B | 16 | – | 1 (<0.25) | – | 4 (0.5) | – | 32 (2) |
| | Ecc20190808 | ST66 | BAL | blaCTX-M-15, blaOXA-1, blaTEM-1B | 16 | aac(6')-Ib-cr, qnrB1 | 16 (8) | – | 2 (1) | – | 32 (4) |
| | Ecc20190903 | ST66 | BAL | blaCTX-M-15, blaOXA-1, blaTEM-1B | 16 | aac(6')-Ib-cr, qnrB1 | 16 (8) | – | 2 (1) | – | 32 (4) |
| | Ecc20190904 | ST66 | BAL | blaCTX-M-15, blaOXA-1, blaTEM-1B | 16 | aac(6')-Ib-cr, qnrB1 | 8 (4) | – | 2 (1) | – | 32 (4) |
| | Ecc20210401 | ST66 | Stool | blaCTX-M-15, blaOXA-1, blaTEM-1B | 16 | aac(6')-Ib-cr, qnrB1 | 16 (8) | – | 2 (1) | – | 16 (2) |
| | Ecc20200502 | ST568 | Stool | blaCTX-M-15, blaOXA-1 | 16 | aac(6')-Ib-cr, qnrB1 | 8 (2) | – | 2 (0.5) | – | 32 (4) |
| | Ecc20210802 | ST66 | Stool | blaCTX-M-15, blaOXA-1, blaTEM-1B | 16 | aac(6')-Ib-cr, qnrB1 | 16 (4) | – | 4 (1) | – | 16 (2) |
| | Ecc20190702 | ST114 | Stool | blaCTX-M-15, blaOXA-1, blaTEM-1B | 32 | aac(6')-Ib-cr, qnrB1, ParC S80I, GyrA S83L | 256 (64) | – | 4 (1) | catA1 | >256 (128) |
| | Ecc20190501 | ST106 | Stool | blaSHV-12, blaTEM-1B | 32 | aac(6')-Ib-cr | 4 (2) | – | 4 (2) | – | 32 (4) |
| | Ecc20200601 | ST66 | Pus | blaCTX-M-15, blaOXA-1, blaTEM-1B | 64 | aac(6')-Ib-cr, qnrB1 | 16 (8) | – | 2 (0.5) | – | 16 (2) |
| | Ecc20190609 | ST66 | Stool | blaCTX-M-15, blaOXA-1, blaTEM-1B | 16 | aac(6')-Ib-cr, qnrB1 | 16 (8) | – | 4 (1) | – | 32 (4) |
| | Ecc20191003 | ST66 | Stool | blaCTX-M-15, blaOXA-1, blaTEM-1B | 16 | aac(6')-Ib-cr, qnrB1 | 16 (8) | – | 4 (1) | – | 32 (4) |
| | Ecc20190502 | ST114 | BAL | blaCTX-M-15, blaOXA-1, blaTEM-1B | 32 | aac(6')-Ib-cr, qnrB1, ParC S80I, GyrA S83L | 256 (128) | – | 2 (0.5) | catA1 | >256 (64) |
| | Ecc20190701 | ST114 | BC | blaCTX-M-15, blaOXA-1 | 64 | aac(6')-Ib-cr, qnrB1, ParC S80I, GyrA S83L | 128 (64) | – | 2 (0.5) | – | 32 (2) |
| | Ecc20190706 | ST114 | Stool | blaCTX-M-15, blaOXA-1, blaTEM-1B | 32 | aac(6')-Ib-cr, qnrB1, ParC S80I, GyrA S83L | 256 (128) | – | 2 (0.5) | catA1 | >256 (64) |
| *Klebsiella pneumoniae* | Kp20190611 | ST405 | Stool | blaCTX-M-15, blaOXA-1, blaTEM-1B | 32 | aac(6')-Ib-cr,oqxA,oqxB,qnrB1 | 64 (64) | – | 2 (0.5) | – | 32 (2) |
| | Kp20190626 | ST307 | Stool | blaCTX-M-15, blaOXA-1, blaTEM-1B | 16 | aac(6')-Ib-cr,oqxA,oqxB,qnrB1, GyrA S83I, ParC S80I | 128 (128) | – | 4 (1) | – | 64 (8) |
| | Kp20190803 | ST405 | Stool | blaCTX-M-15, blaOXA-1, blaTEM-1B | 16 | aac(6')-Ib-cr,oqxA,oqxB,qnrB1 | 32 (8) | – | 2 (0.5) | – | 128 (16) |

TABLE 1 Origins, sequence types, resistomes, and manual minimum inhibitory concentrations with or without 20 mg/L of PAβN for temocillin, ciprofloxacin, tigecycline, and chloramphenicol in the 31 resistant strains[b] (Continued)

| Species | Study name | Origin | ST | Acquired beta-lactamase | Temocillin[a] | Fluoroquinolones AG + QRDR mutations[c] | Ciprofloxacin[a] (+PAβN) | Tigecycline AG[c] | Tigecycline[a] (+PAβN) | Chloramphenicol AG[c] | Chloramphenicol[a] (+PAβN) |
|---|---|---|---|---|---|---|---|---|---|---|---|
| | Kp20190602 | Stool | ST15 | blaCTX-M-15, blaOXA-1, blaTEM-1B | 32 | aac(6')-Ib-cr,oqxA,oqxB,ParC S80I | 256 (128) | – | 4 (2) | – | 64 (4) |
| | Kp20211001 | Stool | ST307 | blaCTX-M-15, blaOXA-1, blaTEM-1B | 16 | aac(6')-Ib-cr,oqxA,oqxB, qnrB1, GyrA S83I, ParC S80I | 256 (64) | – | 2 (1) | – | 32 (4) |
| | Kp20211005 | BA | ST307 | blaCTX-M-15, blaOXA-1, blaTEM-1B | 16 | aac(6')-Ib-cr,oqxA,oqxB, qnrB1, GyrA S83I, ParC S80I | 256 (128) | – | 2 (1) | – | 32 (4) |
| | Kp20190403 | Stool | ST449 | blaSHV-2 | 32 | oqxA,oqxB | 0.5 (<0.25) | – | 4 (1) | – | 128 (8) |
| | Kp20210510 | BA | ST405 | blaCTX-M-15, blaTEM-1B | 16 | oqxA,oqxB,qnrB1 | 4 (1) | – | 2 (1) | – | 32 (4) |
| | Kp20210808 | Stool | ST20 | blaCTX-M-15, blaOXA-1, blaTEM-1B | 32 | aac(6')-Ib-cr | 1 (0.25) | – | 2 (1) | catB3 | >256 (256) |
| | Kp20190607 | Stool | ST45 | blaCTX-M-15, blaTEM-1B | 64 | oqxA,oqxB | 0.5 (<0.25) | – | 2 (0.5) | catA1 | >256 (128) |
| | Kp20191103 | Stool | ST45 | blaCTX-M-15, blaOXA-1, blaTEM-1B | 16 | aac(6')-Ib-cr,oqxA,oqxB, qnrB1 | 16 (8) | – | 2 (1) | – | 32 (8) |
| | Kp20200201 | Stool | ST392 | blaCTX-M-15, blaOXA-1, blaOXA-4, blaTEM-1B, blaTEM-2 | 64 | aac(6')-Ib-cr,oqxA,oqxB, qnrB1, GyrA S83I, ParC S80I | 128 (128) | – | 2 (0.5) | floR | >256 (128) |
| | Kp20201107 | Stool | ST584 | blaCTX-M-15, blaTEM-1B | 32 | oqxA,oqxB,qnrB1 | 2 (<0.25) | – | 4 (0.5) | – | 64 (8) |
| | Kp20201204 | Stool | ST584 | blaCTX-M-15, blaTEM-1B | 16 | oqxA,oqxB,qnrB1 | 2 (1) | – | 4 (0.5) | – | 64 (2) |
| | Kp20210611 | Stool | ST636 | blaCTX-M-15, blaOXA-1, blaTEM-1B | 32 | aac(6')-Ib-cr,oqxA,oqxB, qnrB1 | 4 (4) | – | 2 (1) | – | 16 (2) |
| | Kp20211201 | Catheter | ST870 | blaSHV-106 | 128 | oqxA,oqxB | 256 (128) | – | 2 (1) | – | 16 (2) |

[a]MICs are presented in milligram per liter.
[b]BAL, broncho-alveolar liquid; BC, blood culture; BA, bronchial aspiration; AG, acquired genes; QRDR, quinolones resistance determining region; PAβN, Phe-Arg-β-naphthylamide.
[c]"–" absence of acquired antimicrobial resistance gene or mutations within QRDR

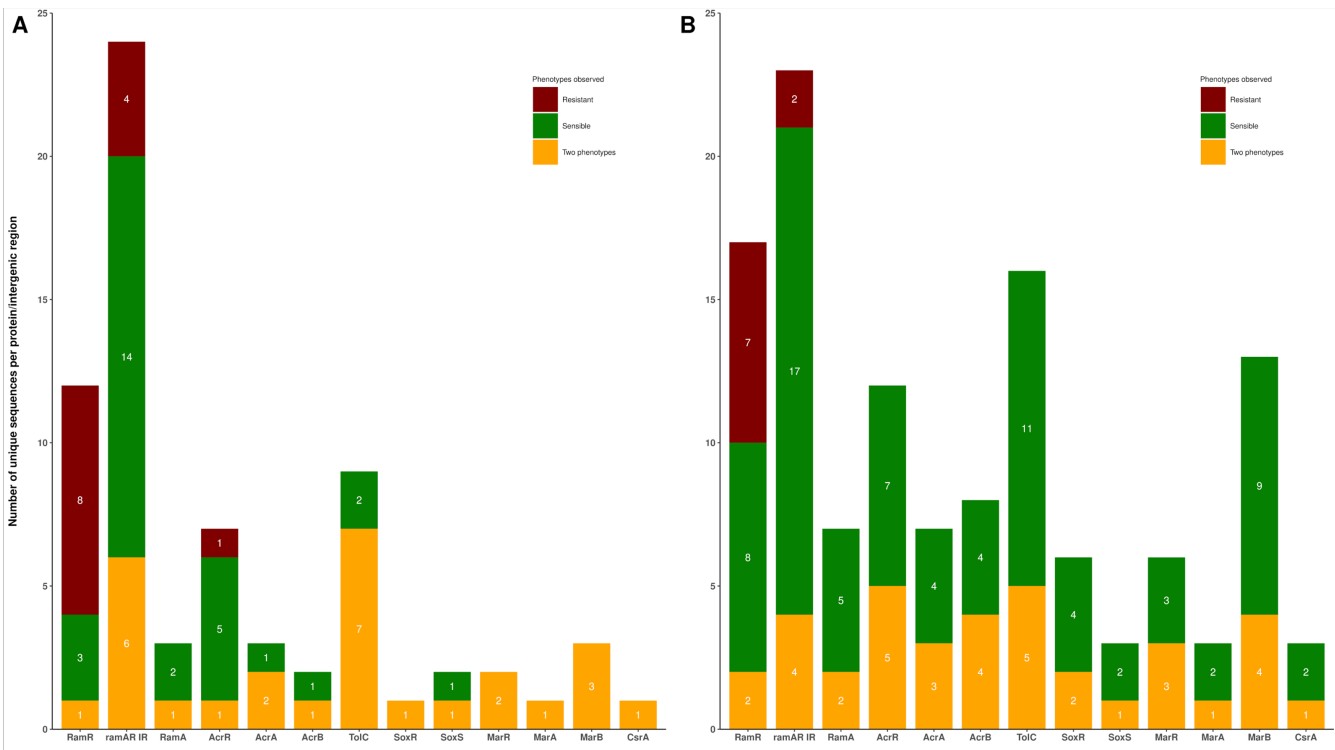

**FIG 1** Diversity of the sequences within the *acrAB-tolC* regulation pathway. The number of unique sequences and the phenotypical association (see Materials and Methods) found in 12 proteins, RamA, RamR, AcrR, AcrA, AcrB, TolC, SoxR, SoxS, MarA, MarB, MarR, and CsrA, and the *ramAR* intergenic region are represented in bar plots. (A) represents the sequence diversity from the genomes of 122 ESBL-producing *K. pneumoniae;* (B) represents the sequence diversity from the genomes of 127 ESBL-producing *E. cloacae* complexes. Sequences only recovered from susceptible strains are represented in green, sequences that were specific to resistant strains are represented in red, and sequences that were found in both susceptible and resistant isolates are colored yellow.

## Correlations between sequences and resistance phenotypes: transcriptomic explorations

*K. pneumoniae* strains with the t2c2 phenotype presented specific mutations within RamR ($n = 8$), *ramAR* IR ($n = 4$), and AcrR ($n = 1$) (Fig. 1A), whereas *Ecc* strains with such phenotype displayed specific modifications within RamR ($n = 7$) and *ramAR* IR ($n = 2$) (Fig. 1B) (see details below). T2c2-resistant strains had associated mutations in other proteins or intergenic regions, which were also found among non-t2c2 stains, suggesting that they did not explain the phenotype observed (Fig. 1).

Among the 15 *Ecc* strains with the t2c2 phenotype, four, which belonged to two different Hoffman *hsp60* clusters (III and IV) and two different STs (ST78 and ST66), had the DNA mutation C148T in the *ramAR* IR (Fig. 2A). This single-nucleotide polymorphism occurred within the RamR DNA-binding site of *ramAR* IR, which likely affects the ability of RamR to play its repressor role (Fig. 2A; Table 2). This hypothesis was supported by qRT-PCR analyses that found overexpression of *ramA* [fold change (FC) = 7.9], *acrA* (FC = 3.5), and *tolC* (FC = 4.5) (Table 3).

Six strains showed several amino acid replacements within RamR. Three ST114 strains had a V39A substitution, which led to a steric modification inside the DNA-binding region of RamR (Table 2; Fig. 3A). Two ST66 strains had substitutions in the first eight N-terminal amino acids, leading to two steric hindrances, three side chain polarity modifications, and three charge modifications (Fig. 3A). These substitutions are likely to change the secondary and tertiary structures of the N-terminal part of RamR, which is the DNA-binding region of the protein. Lastly, one ST66 strain had a F155L amino acid substitution (Table 2; Fig. 3A). This 155 position is known to be one of the major sites for protein–substrate interactions. Substitution of a large and aromatic amino acid by a

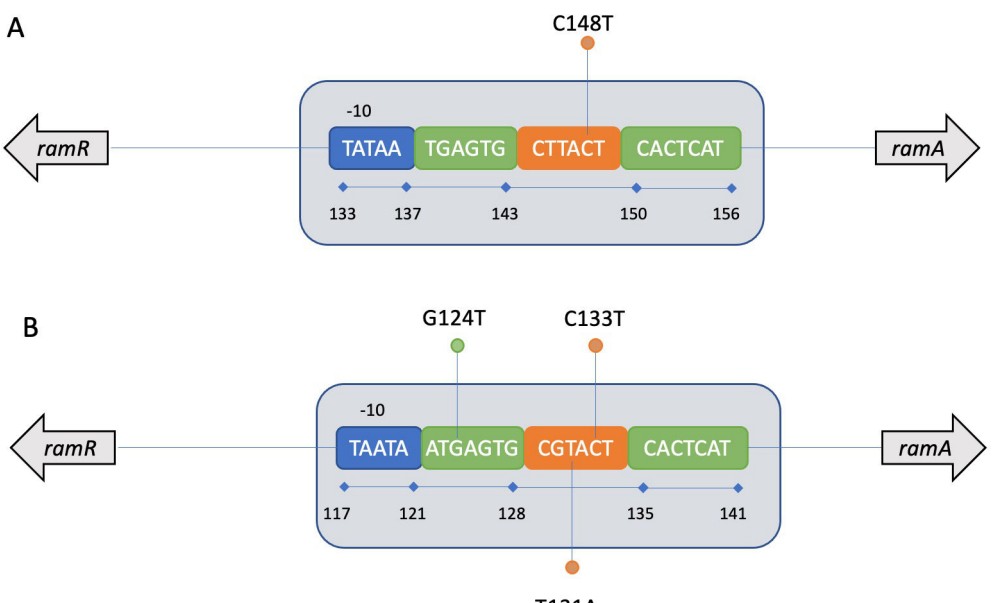

**FIG 2** Organization of the RamR DNA-binding domain found among *Enterobacter cloacae* complex (A) and *Klebsiella pneumoniae* (B) strains. Annotation of the region was performed according to the description of Baucheron et al. (16). −10 Tata boxes are represented in blue, nucleotide binding sites are represented in green, and the six nucleotides' "linkers" are in orange. Nucleotide mutations retrieved among t2c2 strains are represented by circles.

smaller and nonaromatic amino acid should change the conformation of RamR. All these amino acid substitutions led to an overexpression of the genes *ramA* (FC min = 9.7, FC max = 10.5), *acrA* (FC min = 3.6, FC max = 4.8), and *tolC* (FC min = 1.2, FC max = 5.9) (Table 3).

Five *Ecc* strains showed a deletion of the N-terminal part of RamR (Table 2; Fig. 3B through D). One ST66 strain had a 1_35del mutation in addition to eight amino acid substitutions within the DNA-binding site (Fig. 3B). One ST106 strain had a 29_32del (Table 2; Fig. 3C), whereas three isolates from different STs (ST568, ST66, and ST114) had a 1_69del (Table 2; Fig. 3D). All these deletions completely or partially destroyed the DNA-binding region of RamR, which most likely led to an inability of the protein to be a transcriptional regulator. The transcriptomic analyses indicated an overexpression of the three genes controlled by the *ramAR* locus: *ramA* (FC min = 9.7, FC max = 12), *acrA* (FC min = 1.4, FC max = 5.3), and *tolC* (FC min = 3.9, FC max = 5.2) (Table 3). Note that for the strain Ecc20190702, no dysregulation of the transcription of *tolC* has been found despite overexpression of *ramA* (Table 3).

Among the 16 *K. pneumoniae* strains with the t2c2 phenotype, five presented a mutation inside the *ramAR* IR, while 10 had mutations within the RamR sequence (Table 2; Fig. 2 and 4). One strain, Kp20210808, had, in addition to the mutation in RamR, another specific mutation inside AcrR amino acid sequence A80S, which led to the replacement of a small and nonpolar amino acid by a larger and polar one. For the strain Kp202106011, no explanation could be provided by the genomic explorations despite a t2c2 phenotype with an obvious decrease of MICs by the addition of PAβN (Tables 1 and 2).

Inside the *ramAR* IR, two strains belonging to the STs ST392 and ST45 had, respectively, T131A and C133T nucleotide substitutions, which both occurred within the RamR DNA-binding region (Table 2; Fig. 2B). In addition, two ST584 strains had a G124T substitution, which changed the "ATGAGTT-N6-GGTCGAT" sequence inside the RamR binding site (Table 2; Fig. 2B). Together, these substitutions could modify the DNA-RamR interaction, likely leading to a weaker RamR repressor activity of efflux pumps encoding genes and consequently a higher antimicrobial resistance. On the other hand, the strain

**TABLE 2** Sequence types of the resistant strains, RamR- and *ramAR* intergenic region-associated mutations, and biological significance

| Study name | ST | RamR mutations associated with resistant strains[a] | *ramAR* intergenic region mutations associated with resistant strains[a] | Biological significance of genomic/protein modifications |
|---|---|---|---|---|
| Ecc20200803 | ST78 | – | C148T | Mutations within the RamR DNA-binding region: linker |
| Ecc20190808 | ST66 | | | |
| Ecc20190903 | ST66 | | | |
| Ecc20190904 | ST66 | | | |
| Ecc20210401 | ST66 | p1_35del, p.N36M, A37R, G38A, V39L, A40L, E41K, T43H, L44C | – | 1. Loss of the DNA-binding domain<br>2. Amino acid modifications: steric hindrance, side chain polarity modifications, charge modification |
| Ecc20200502 | ST568 | 1_69del | – | Loss of the DNA-binding domain |
| Ecc20210802 | ST66 | | | |
| Ecc20190702 | ST114 | | | |
| Ecc20190501 | ST106 | 29_32del | – | Loss of the DNA-binding domain |
| Ecc20200601 | ST66 | F155L | – | Amino acid modifications: steric hindrance inside a key position for protein–substrate interactions |
| Ecc20190609 | ST66 | V1W, A2H, R3V, P4R, K5R, S6V, E7K, D8I | – | Amino acid modifications: steric hindrance, side chain polarity modification, charge modification |
| Ecc20191003 | ST66 | | | |
| Ecc20190502 | ST114 | V39A | – | Amino acid modifications: steric hindrance inside the DNA-binding domain |
| Ecc20190701 | ST114 | | | |
| Ecc20190706 | ST114 | | | |
| Kp20190611 | ST405 | Y92N | – | Amino acid modifications: steric hindrance |
| Kp20190626 | ST307 | Y47N | – | Amino acid modifications: steric hindrance |
| Kp20190803 | ST405 | W89R | – | Amino acid modifications: steric hindrance, charge modification |
| Kp20190602 | ST15 | L154Q | – | Amino acid modifications: steric hindrance and polarity modification of the side chain inside a key position for protein–substrate interactions |
| Kp20211001 | ST307 | K9I | – | Amino acid modifications: steric hindrance, charge modification |
| Kp20211005 | ST307 | | | |
| Kp20190403 | ST449 | 130_193del | – | Loss of C-terminal extremity of RamR, no possibility for dimerization |
| Kp20210510 | ST405 | 122_193del | – | Loss of C-terminal extremity of RamR, no possibility for dimerization |
| Kp20210808 | ST20 | | | |
| Kp20190607 | ST45 | 101_193del | – | Loss of C-terminal extremity of RamR, no possibility for dimerization |
| Kp20191103 | ST45 | No specific mutation | C133T | Mutations within the RamR DNA-binding region: linker |
| Kp20200201 | ST392 | No specific mutation | T131A | Mutations within the RamR DNA-binding region: linker |
| Kp20201107 | ST584 | No specific mutation | G124T | Mutation inside the palindromic sequence ATGAGTG within the RamR DNA-binding region |
| Kp20201204 | ST584 | | | |
| Kp20211201 | ST870 | No specific mutation | G1298T | Unknown significance |
| Kp202106011 | ST636 | No specific mutation | No specific mutation | No explanation |

[a]- absence of mutation within the region of interest

Kp20211201 had a unique substitution within the *ramAR* IR, G1298T, whose impact on molecular mechanisms is unknown (Table 2). All the nucleotides' substitutions within the *ramAR* region changed the expression of *ramA* (FC min = 1.2, FC max = 13), *acrA* (FC min = 1.7, FC max = 4.6), and *tolC* (FC min = 3, FC max = 6.7), respectively (Table 3).

Among the 10 *K. pneumoniae* strains with specific RamR mutations, three ST114 strains, one ST405, one ST307, and another ST405 showed V39A, Y92N, Y47N, and W89R amino acid substitutions, respectively (Table 2; Fig. 4A). All these alterations led to steric hindrance inside the DNA-binding domain (Fig. 4A). It should be noted that the W89R was also responsible for the replacement of a neutral amino acid, tryptophan, by a positive amino acid, arginine, which may have consequences for the secondary

**TABLE 3** Transcriptomic consequences of the modifications of the *ramAR* intergenic region or the RamR sequences regarding three genes controlled by the ramAR locus: *ramA*, *acrA,* and *tolC*

| Species | Genomic modifications | | | qRT-PCR (mean ± standard deviations) | | |
|---|---|---|---|---|---|---|
| | Localization | Nature | Strain used | *ramA* | *acrA* | *tolC* |
| *Enterobacter cloacae* complex | *ramAR* intergenic region | C148T | Ecc20190808 | 7.9 ± 0.8 | 3.5 ± 1.0 | 4.5 ± 0.8 |
| | DNA-binding domain | 1_69del | Ecc20190702 | 12 ± 1.1 | 5.3 ± 1.2 | −0.15 ± 1.3 |
| | | 1_35del & multiple AA substitutions | Ecc20210401 | 9.7 ± 0.9 | 3.6 ± 0.7 | 5.2 ± 1.0 |
| | | 29_32del | Ecc20190501 | 9.9 ± 0.8 | 1.4 ± 2.7 | 3.9 ± 0.6 |
| | | V1W. A2H. R3V. P4R. K5R. S6V. E7K. D8I | Ecc20190609 | 10.3 ± 2.3 | 4.8 ± 0.2 | 5.9 ± 1.8 |
| | | V39A | Ecc20190502 | 10.5 ± 1.6 | 4.3 ± 0.5 | 1.2 ± 2.9 |
| | Position associated with protein–substrate interactions | F155L | Ecc20200601 | 10.2 ± 3.6 | 5.3 ± 2.8 | 6.1 ± 3.4 |
| *Klebsiella pneumoniae* | *ramAR* intergenic region | G124T | Kp20201204 | 13 ± 1.3 | 4.6 ± 2.1 | 6.7 ± 2.2 |
| | | T131A | Kp20200201 | 1.2 ± 1.3 | 2.1 ± 2.3 | 3 ± 2.1 |
| | | C133T | Kp20191103 | 2.3 ± 1.4 | 1.7 ± 3.2 | 3.6 ± 2 |
| | | G1298T | Kp20211201 | 4.4 ± 2.3 | 3 ± 4.1 | 4.1 ± 3.1 |
| | DNA-binding domain | K9I | Kp20211005 | 7.3 ± 1.8 | 5.6 ± 2.4 | 5.5 ± 2.1 |
| | | Y47N | Kp20190626 | 1.9 ± 1.4 | 2.8 ± 1.4 | 3.2 ± 1.8 |
| | Position associated with protein–substrate interactions | W89R | Kp20190803 | 4.5 ± 1.4 | 2.5 ± 3.7 | 2.2 ± 3.6 |
| | | Y92N | Kp20190611 | 2.5 ± 1.1 | 2.9 ± 2.2 | 4.5 ± 2.2 |
| | | L154Q | Kp20190602 | 3.5 ± 1.7 | 3.8 ± 3.2 | 3.3 ± 3 |
| | Loss of C-terminal extremity of RamR | 130_193del | Kp20190403 | 13 ± 1.7 | 10 ± 3.3 | 9 ± 3 |

and tertiary structures of the protein (Table 2; Fig. 4A). The strain Kp20190602 (ST15) displayed a L154Q substitution that caused steric hindrance and polarity modification of the side chain of the amino acid inside a key position for protein–substrate interactions (Table 2; Fig. 4A).

Finally, four strains of *K. pneumoniae* belonging to ST449, ST405, ST20, and ST45 harbored important deletions in the C-terminal part of RamR, 130_193del, 122_193del, 122_193del, and 101_193del, respectively (Table 2; Fig. 4B through D). All these deletions have in common the disappearance of the ninth alpha helix, which allows the dimerization of RamR; thus, RamR is likely to not be functional. The modifications of the DNA-binding domain, the alteration of the key positions of the protein–substrate interaction, or the massive C-terminal deletion of RamR, all these protein rearrangements caused a *ramA* (FC min = 1.9, FC max = 13), *acrA* (FC min = 2.8, FC max = 10), and *tolC* (FC min = 2.2, FC max = 9) overexpression (Table 3).

It is noteworthy that some genomic alterations led to a stronger dysregulation of *ramA*, *acrA,* and *tolC* than others, suggesting an unequal consequence on the integrity and functionality of RamR.

## DISCUSSION

Mutations in RamR and AcrR or within the *ramAR* IR associated with AcrAB-TolC dysregulation and antimicrobial resistance have been described in many species (4, 7, 8, 17). In this work, more than 10% of the ESBL-*Ecc* and ESBL-*K. pneumoniae* strains isolated in ICUs for 3 years showed a t2c2 phenotype, which was associated with mutations inside the *ramAR* locus. These mutations were found within two species, in 6 and 10 different STs among ESBL-*Ecc* and *ESBL-K. pneumoniae,* respectively.

Bioinformatics analyses of genes linked to the expression of RND pumps revealed that sequence diversity was limited to a few elements. The total number of unique sequences was higher in *Ecc* than in *K. pneumoniae*, which could be explained by the existence of several species merged into the *E. cloacae* complex whereas only *K. pneumoniae* strains were found in our study. In both species/complexes, sequences of CsrA, MarA, SoxS, SoxR, RamA, and MarR were conserved, whereas AcrR, *ramAR* IR, and RamR showed

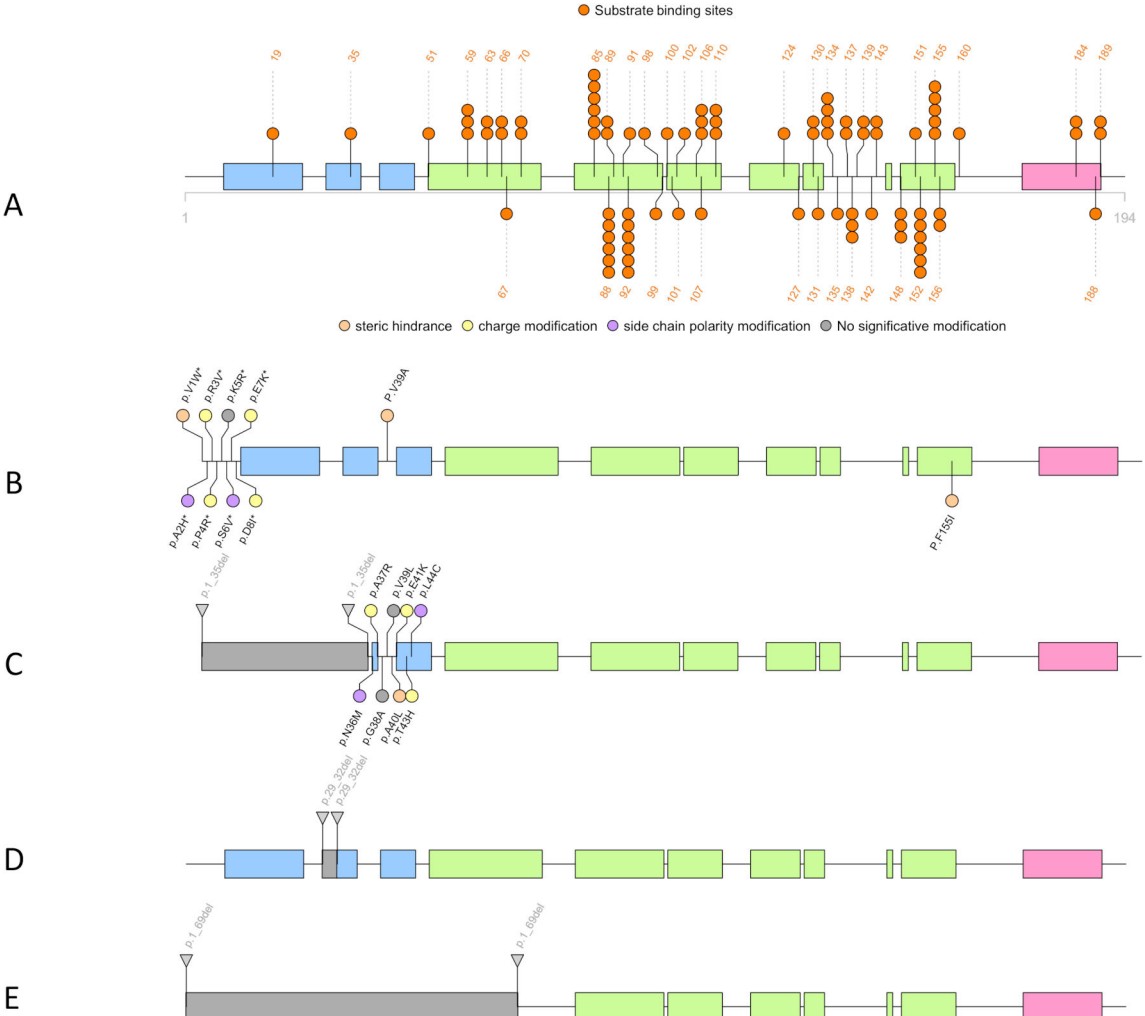

**FIG 3** Illustration of RamR associated with *Enterobacter cloacae* complex t2c2 phenotype. Blue rectangles represent the HTH domain of the RamR. Green rectangles constitute the regions involved in the protein–substrate interactions. Pink rectangles are the ninth alpha helices, which are involved in dimerization with a second RamR protein. (A and B) Amino acid substitutions associated with resistant *Enterobacter cloacae* complex strains are represented in circles. Consequences of the substitution are organized by colors: steric hindrance (orange), amino acid charge modification (yellow), side chain polarity modification (purple), and no significative modification (gray). (B–D) represent the N-terminal deletions found in several RamR. For each individual modification of RamR, the sequence type of the strains as well as the total of isolates is indicated near the blue boxes.

the highest proportion of mutations. Consequently, *csrA, marA, soxS, soxR, ramA,* and *marR* could be considered as "stable" genes, probably because of their impact on several crucial cellular processes. Indeed, CsrA plays a major role in intracellular carbon metabolism by controlling several essential enzymes involved in glucose metabolism in *E. coli* (18). *marRAB* and *soxRS* operons are both involved in oxidative stress responses (10, 19). It has been demonstrated that for clinical *E. coli* strains, the selection pressure constrains the mutations within the *marR* gene due to important fitness costs (20). Moreover, mutations within the *soxRS* operon have been described among *E. coli* and *K. pneumoniae* strains; they lead to a cross-antimicrobial resistance phenotype (15, 21).

At the same time, it has been described that *ramRA* has pleiotropic effects on the cell, such as modification of lipid A synthesis, antimicrobial susceptibility, and macrophage interactions (22). Therefore, the fitness cost for the cell of mutations within the *ramRA* operon is probably lower than those within *marRAB*, *soxRS*, and *csrA,* allowing a higher sequence diversity in this genomic region. The *ramAR* operon seems to be an important adaptive genetic element for the *Ecc* and *K. pneumoniae* strains. This statement was

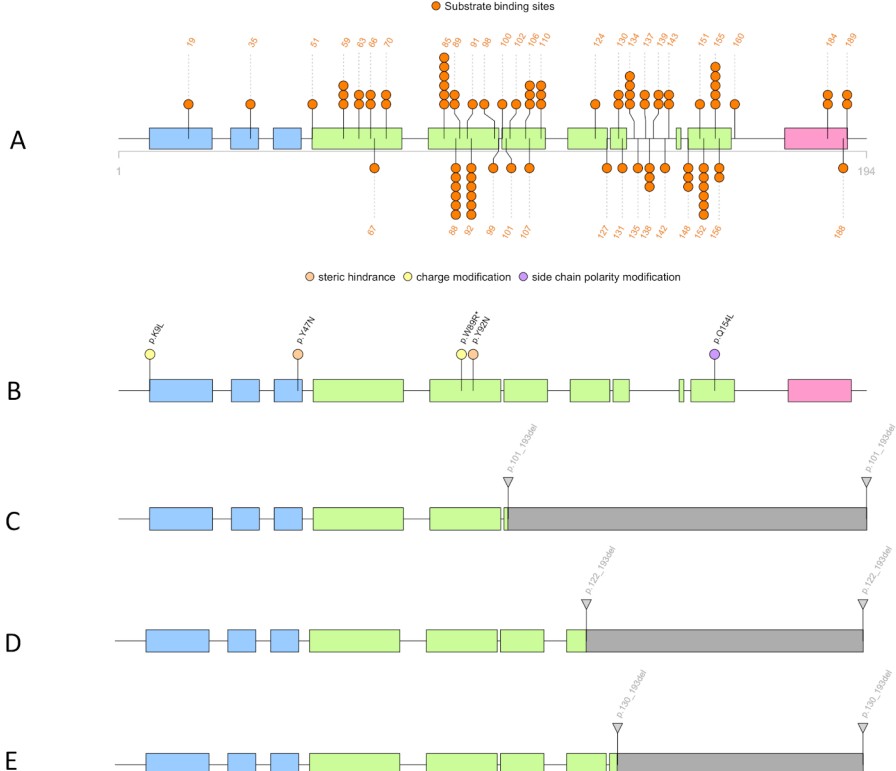

**FIG 4** Illustration of RamR associated with *Klebsiella pneumoniae* t2c2 phenotype. Blue rectangles represent the HTH domain of the RamR. Green rectangles constitute the regions involved in the protein–substrate interactions. Pink rectangles are the ninth alpha helices, which are involved in dimerization with a second RamR protein. (A) Amino acid substitutions associated with resistant *Klebsiella pneumoniae* strains are represented in circles. Consequences of the substitution are organized by colors: steric hindrance (orange), amino acid charge modification (yellow), and side chain polarity modification (purple). (B–D) represent the C-terminal deletions found in several RamR. For each individual modification of RamR, the sequence type of the strains as well as the total of isolates is indicated near the blue boxes. *One ST405 strain had the W89R substitution while another one had the Y92N.

also supported by the fact that no *E. coli* t2c2 strain was found and that isolates of this species did not possess the *ramAR* locus.

It should be noted that all the resistant strains harbored an altered sequence of the *ramAR* IR or RamR but not both. This is in accordance with a study of the *ramAR* locus among *Salmonella* Typhimurium strains, which also concluded that cells had specific mutations in one or another region but not simultaneously (17).

Three types of modifications have been found in mutated RamR. First, the DNA-binding site could be altered by (i) massive deletions, as it has been found in *Ecc* strains in other studies (4, 23, 24), and (ii) amino acid substitutions. These last have been reported in several species, but their localization can differ (7, 17, 25). It should be noted that the tryptophan substitution at position 89 has also been described in a *K. pneumoniae* strain (25). In accordance with previously published studies, most of the mutated RamR had a unique amino acid substitution (7, 25). Deletions and/or amino acid substitutions in the RamR DNA-binding domain could lead to a weaker or complete abolition of its transcriptional inhibitory function, resulting in *ramA* and RND pump overexpression (4). The second type of RamR alteration was, as previously described (7), the deletion of the C-terminal part of the protein, which resulted in the absence of the ninth alpha helix involved in the dimerization of the protein (12). Third, two proteins harbored amino acid substitutions in positions 154 and 155. These last have been described as key positions for protein–substrate interactions in *Salmonella enterica* serovar Typhimurium (12). Amino acid replacement inside the 155 position or near it (154) could change the

behavior of the protein, which will result in lower functionality. Furthermore, deletions within positions 154 and 155 of the RamR of a *K. aerogenes* strain were associated with high resistance to chloramphenicol (23).

Mutations associated with resistant strains have also been found inside the *ramAR* IR. In both *Ecc* and *K. pneumoniae*, the RamR DNA-binding region was subjected to nucleotide substitutions that probably interfered with the *ramA* and/or *ramR* expression and consequently modified the antimicrobial resistance (Fig. 2).

For two *K. pneumoniae* strains, there was a modification in the "ATGAGTT-N6-GGTCGAT" sequences corresponding to the RamR binding motif (Fig. 2) (26). The overexpression of AcrAB-TolC after modification of this protein binding sequence was described in previous studies (17, 27). Four *Ecc* and two *K. pneumoniae* strains showed nucleotide substitutions within the "N6-linker" nucleotides. A study performed with *Salmonella enterica* serovar Typhimurium highlighted that two RamR homodimers bind the DNA, one per DNA strand. The DNA-binding site of RamR consists of a 28-bp nucleotide sequence including the −10 region binding site of the RNA polymerase as well as the "ATGAGTT-N6-GGTCGAT" sequence (16). Interestingly, it has been shown in *Salmonella enterica* that a 2-bp deletion including one nucleotide inside the linker alters RamR-DNA interactions, leading to the overexpression of *ramA*, *acrB*, and *tolC* by 6.6-, 3.4-, and 2.5-fold, respectively (16). These results are in the same range as those obtained in this study.

It has been demonstrated that RamA controls not only the expression of *acrAB-tolC* but also other pump-encoding genes such as *oqxAB* and *yrbB-F* (22). It also controls the lipid A moiety by modulating the transcription of *lpxO*, *lpxC,* and *lpxL-2*. Therefore, RamA-overexpressing strains are less susceptible to polymyxin, colistin, and cationic antimicrobial peptides due to membrane modifications and are more resistant to macrophage phagocytosis, which conferred better systemic diffusion to the lung and spleen in a mouse model of infection (22). All these characteristics provide selective advantages for clinical bacterial strains.

Finally, it has been demonstrated that *acrAB*-overexpressing cells of *E. coli* and *S.* Typhimurium do not express *mutS* (28). Thus, the lack of DNA mismatch repair led to hypermutable phenotypes (28). It could then be hypothesized that AcrA-TolC overexpression through RamR or *ramAR* IR mutations allows bacterial strains to acquire chromosomal mutations, leading to a heterogeneous bacterial population among which some cells could better adapt to stress. According to this evolutionary point of view, in *K. pneumoniae* and *Ecc,* the *ramRA* locus seems to be a major genetic element of adaptation for bacteria, in particular in decreasing its susceptibility to antibiotics.

## Conclusion

Almost 7% of ESBL-*Enterobacterales* isolated in ICUs showed resistance to TEM, CIP, TIG, and CHL named t2c2 phenotype in the present study. The bacterial species were exclusively *Enterobacter hormaechei* and *Klebsiella pneumoniae*, and no t2c2 phenotype was found among *E coli*. For 30/31 strains, there was a clear correlation between the alteration of the sequences in the *ramAR* operon, the t2c2 phenotype, and overexpression of *ramA*, *acrA,* and *tolC*. Sequence diversity and transcriptomic analyses demonstrated that RamR and the *ramAR* IR played key roles in the acquisition of antimicrobial resistance and bacterial adaptation among clinical strains. Further explorations are needed to confirm these results in a multicentric and prospective study.

## MATERIALS AND METHODS

### Bacterial strains

In the Teaching Hospital of Caen, patients from medical, surgical, or neonatology ICUs were screened weekly for ESBL-E carriage using rectal swabs and selective media (Chromid BLSE, Biomérieux, Marcy l'Etoile, Auvergne-Rhône-Alpes, France). Identification

at the species level was performed using a matrix-assisted laser desorption ionization–time-of-flight mass spectrometry method (Bruker, Billerica, MA, USA). The presence of an ESBL-encoding enzyme was confirmed using the combination disk test according to the European Committee on Antimicrobial Susceptibility Testing guidelines (https://eucast.org/). Between 2019 and 2021, the first ESBL-E strains identified as *E. coli*, *K. pneumoniae,* and *Ecc* found in clinical or screening samples from each patient were gathered for whole-genome sequencing (WGS) and further antimicrobial susceptibility characterization.

## Antimicrobial susceptibility tests

MICs were determined using the FRAMgGN plate of the Sensititre Antimicrobial Susceptibility Testing System (ThermoFisher Scientific, Waltham, MA, USA) according to the manufacturer's instructions. The antimicrobial agents or combinations tested included amikacin, aztreonam, cefepime, ceftazidime, CIP, colistin, ertapenem, ceftolozane-tazobactam, gentamicin, imipenem, meropenem, piperacillin-tazobactam, TEM, TIG, and sulfamethoxazole. MIC values were determined using the Sensititre Vizion Digital MIC Viewing System (ThermoFisher Scientific, Waltham, MA, USA). Clinical interpretation was performed according to the guidelines of the "*Comité de l'Antibiogramme de la Société Française de Microbiologie*" CA-SFM EUCAST 2020 (29).

## Selection of bacterial strains of interest and MIC determination

Strains that demonstrated a decreased susceptibility to CIP (>0.5 mg/L), TIG (>0.5 mg/L), and TEM (>8 mg/L) were selected for further investigations. MICs for TIG, TEM, CIP, and CHL were determined by the broth microdilution method carried out using Mueller–Hinton (MH) broth in biological triplicates at a temperature of 35℃ ± 2℃. The bacterial suspensions used to inoculate microplates were made from 0.5 McFarland suspensions that were then diluted to 100th in cation-adjusted MH broth.

As controls, *Staphylococcus aureus* ATCC 29213 was used for CIP, CHL, and TIG, and the *Enterobacter hormaechei* FY01 strain was used for TEM (4).

The impact of the RND efflux pump inhibitor PAβN (Sigma-Aldrich, Saint Louis, MO, USA) at 20 mg/L on the MIC values was tested for CIP, CHL, and TIG (26).

MICs for TEM were not tested in the presence of PAβN, as the MICs observed are not linked to RND pump dysregulation but rather to a modification of *ompF* expression, as has already been mentioned (4). The t2c2 phenotype was defined as a resistance to four antibiotics CIP (>0.5 mg/L), TIG (>0.5 mg/L), TEM (>8 mg/L), and CHL (>8 mg/L).

## DNA sequencing and strain typing

High-throughput WGS was carried out at the *Platforme de Microbiologie Mutualisée* (P2M) from the Pasteur International Bioresources network (PIBNet; Institut Pasteur, Paris, Île de France, France). DNA extraction was performed using the MagNAPure 96 system (Roche, Bâle, Swiss). Libraries were prepared using the Nextera XT kit (Illumina, San Diego, CA, USA), and sequencing was performed with the NextSeq 500 system (Illumina), which generated 150 bp paired-end reads. Alien Trimmer software was used for read trimming and clipping. The quality of the filtered fastq files was evaluated using fastqc software. Genomes were *de novo*-assembled using SPAdes v. 3.12 software. The quality of the genome assembly was assessed using the Quast tool (30). Species of the sequenced bacteria were verified using the rMLST program (31). STs were determined using appropriate multilocus sequence typing schemes (32–34). Resistomes of the strains were determined using the Resfinder database with 90% identity and coverage parameters (35).

## Gene sequence extraction and sequence clustering

Nucleic sequences of 13 elements involved within the *acrAB-tolC* regulation pathway were studied. In total, 12 genes, *ramA, ramR, acrR, acrA, acrB, tolC, soxR, soxS, marA, marB, marR,* and *csrA*, and one chromosomal region located between *ramR* and *ramA* were extracted from the genome of each strain using blastn with megablast parameters.

Reference sequences used for the blastn database were extracted from NZ_CP012165.1 *Enterobacter hormaechei* subsp. *xiangfangensis* strain 34978 chromosome complete genome and CP052181.1 *Klebsiella pneumoniae* strain F16KP0037 chromosome complete genome. Both genomes were chosen because of the high-quality sequencing methods using long-read sequencing.

Corresponding peptide sequences were obtained from extracted nucleic sequences using the algorithm Prodigal (32). The sequences of RamA, RamR, AcrR, AcrA, AcrB, TolC, SoxR, SoxS, MarA, MarB, MarR, and CsrA were compared and clustered using the cd-hit program (36). The nucleotide sequences of the *ramAR* IR were compared and clustered using the cd-hit-est program (36). All cd-hit analyses were performed using 100% identity and length thresholds.

## Selection of sequence alterations in the acrAB-tolC regulation pathway

Protein and nucleotide regions with sequences that were only recovered from resistant strains were further analyzed. To consider the protein/nucleotide region, all unique sequences were aligned using muscle (37). Alignments were browsed using AliView (38) to gather the protein/nucleotide modifications that were associated with resistant strains. Then, the biological significance of these differences was tracked using previously published data and protein databases such as UniProt (39), RCSB PDB (40), and InterPro (41). Reference sequences chosen for amino acid or nucleotide substitution studies were those from susceptible strains (Kp20200502 and Ecc20200301 for *K. pneumoniae* and *Ecc* complex strains, respectively).

## RNA extraction and RT-qPCR

Total RNAs were extracted and quantified from bacterial cells grown to the late exponential phase using the ARN Maxwell RSC miRNA Tissue Kit and the Maxwell instrument (Promega, Madison, WI, USA). cDNA synthesis from total RNAs (0.1 µg) and RT-qPCR were performed using a Platinum SYBR Green One-Step qRT-PCR Kit (Invitrogen, Waltham, MA, USA) according to the manufacturer's instructions. The expression ratios of *ramA*, *acrA,* and *tolC* in strains with a t2c2 phenotype were determined by comparison with the transcription levels of the *K. pneumoniae* strain from our lab P013-01-S1, which had a wild-type antimicrobial resistance phenotype profile. The genome of the strain and its antibiogram are available under the following accession number: SAMN41383518. The expression of 23S ribosomal RNA was used as an internal control. Experiments were conducted at least three times. The primers used are available in Table S4.

## Statistical analyses

Statistical analyses were performed using pandas version 0.24.1 (https://pandas.pydata.org/) and scipy version 1.2.1. Categorical variables were compared using the Chi$^2$ test. A *P*-value inferior to 0.05 was considered as significant.

### ACKNOWLEDGMENTS

We thank Michel Auzou, Guillaume Leduc, Isabelle Renouard, Anaïs Rousseau, and Mamadou Godet for their excellent technical assistance. Part of this work was performed on computing resources provided by CRIANN (Normandy, France).

The 3-year PhD program of François Gravey was funded by "Région Normandie."

## AUTHOR AFFILIATIONS

[1]Department of Infectious Agents, Bacteriology, Université de Caen Normandie, Univ Rouen Normandie, INSERM, Normandie Univ, DYNAMICURE UMR 1311, CHU Caen Normandie, Caen, France

[2]Univ de Caen Normandie, Univ Rouen Normandie, INSERM, DYNAMICURE UMR 1311, Caen, France

[3]Service de Réanimation Chirurgicale, Univ de Caen Normandie, CHU de Caen Normandie, Caen, France

[4]Service de Réanimation Médicale, Univ de Caen Normandie, CHU de Caen Normandie, Caen, France

[5]Service de Réanimation Néonatale, Univ de Caen Normandie, CHU de Caen Normandie, Caen, France

[6]Service de Réanimation Pédiatrique, Univ de Caen Normandie, CHU de Caen Normandie, Caen, France

[7]Service de Bactériologie, CHU de Rennes Pontchaillou, Rennes, France

## AUTHOR ORCIDs

François Gravey  http://orcid.org/0000-0001-8672-8286
Jean-Christophe Giard  http://orcid.org/0000-0001-8588-2732

## AUTHOR CONTRIBUTIONS

François Gravey, Conceptualization, Formal analysis, Investigation, Methodology, Software, Validation, Writing – original draft, Writing – review and editing | Alice Michel, Formal analysis, Investigation, Resources, Validation, Writing – original draft, Writing – review and editing | Bénédicte Langlois, Data curation, Writing – original draft, Writing – review and editing | Mattéo Gérard, Investigation, Resources | Sébastien Galopin, Investigation, Methodology, Resources, Writing – original draft, Writing – review and editing | Clément Gakuba, Investigation, Writing – original draft, Writing – review and editing | Damien Du Cheyron, Investigation, Writing – original draft, Writing – review and editing | Laura Fazilleau, Investigation, Writing – original draft, Writing – review and editing | David Brossier, Data curation, Investigation, Methodology, Visualization, Writing – original draft, Writing – review and editing | François Guérin, Conceptualization, Data curation, Investigation, Methodology, Supervision, Validation, Visualization, Writing – original draft, Writing – review and editing | Jean-Christophe Giard, Conceptualization, Data curation, Investigation, Resources, Supervision, Validation, Visualization, Writing – original draft, Writing – review and editing | Simon Le Hello, Conceptualization, Data curation, Investigation, Methodology, Resources, Validation, Visualization, Writing – original draft, Writing – review and editing

## DATA AVAILABILITY

*In silico* analyses were performed using publicly accessible and published algorithms/software. The parameters used are detailed in Materials and Methods. The sequences of the strains are available at BioProject PRJNA876630. Note that genomes of the strains Kp20200902, Kp20200907, Kp20201104, Kp20201003, Kp20201101, Kp20201103, Kp20201102, Kp20191002, Kp20201107, Kp20201201, and Kp20201204 are available following this BioProject accession number PRJNA811107.

## ETHICS APPROVAL

This study was conducted in compliance with the Helsinki Declaration (ethical principles for medical research involving human subjects) and in accordance with the guidelines of the research board of our teaching hospital in Caen, France. This was a noninterventional study: the specimens used in this study were part of the routine patient management without any additional sampling. No objection was provided by patients or their families.

## ADDITIONAL FILES

The following material is available online.

### Supplemental Material

**Supplemental material (Spectrum03548-23-s0001.xlsx).** Tables S1 to S4.

### Open Peer Review

**PEER REVIEW HISTORY (review-history.pdf).** An accounting of the reviewer comments and feedback.

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
