## [Reviewer comments · Microbiology Spectrum]

Microbiology Spectrum

Central role of the *ramAR* locus in the multidrug resistance in ESBL *Enterobacterales*

Francois Gravey, Alice Michel, Bénédicte Langlois, Mattéo Gérard, Sébastien Galopin, Clement Gakuba, Damien du Cheyron, Laura Fazilleau, David Brossier, François Guérin, Jean-Christophe Giard, and Simon Le Hello

Corresponding Author(s): Francois GRAVEY, Universite de Caen Normandie

Review Timeline:

Submission Date:	October 3, 2023
Editorial Decision:	November 16, 2023
Revision Received:	February 28, 2024
Editorial Decision:	April 1, 2024
Revision Received:	May 15, 2024
Accepted:	May 17, 2024

Editor: Shannon Manning

Reviewer(s): The reviewers have opted to remain anonymous.

Transaction Report:

DOI: <https://doi.org/10.1128/spectrum.03548-23>

Re: Spectrum03548-23 (Central role of the *ramAR* locus in the multidrug resistance in ESBL *Enterobacteriales*)

Dear Dr. Francois GRAVEY:

Thank you for the privilege of reviewing your work, which has now been reviewed by 2 experts in your field. Both have suggested important revisions in order for your manuscript to be reconsidered for publication. Reviewer 1 has highlighted the need to include additional references and statements that dampen some of your claims, while Reviewer 2 has indicated the need for performing some additional experiments and re-analyzing the data. Inclusion of *acrAB* expression data, for instance, would support the claim about specific impacts of some mutations. There are also concerns about the presentation of the manuscript, which could be enhanced by additional rounds of edits and better organization of the different sections. Similarly, Reviewer 2 pointed out issues and inconsistencies with the supplementary data. Keeping these and the remaining comments in mind, we would like to invite you to submit a revision for consideration. The complete list of reviewer comments are included below.

If you choose to resubmit, then please be sure that your BioProject sequences are made publicly available prior to submission and return the manuscript within 60 days. If you cannot complete the modifications within this time period, please contact me. If you do not wish to modify the manuscript and prefer to submit it to another journal, then please notify me immediately so that the manuscript may be formally withdrawn from consideration by Spectrum.

Revision Guidelines

Sincerely,
Shannon Manning
Editor
Microbiology Spectrum

Reviewer #1 (Comments for the Author):

Specific Comments concerning the manuscript intitled : « Central rôle of the RamAR locus in the multidrug resistance in ESBL Enterobacterales ». Spectrum03548-23

General comments :

The authors screened 443 ESBL Enterobacteria for efflux resistance phenotype to 4 antibiotics (temocillin, tigecycline, ciprofloxacin, chloramphenicol), and identified those with mutations in the ramAR locus. The RamA regulatory protein is known to be primarily involved in regulating the active efflux of antibiotics in *Klebsiella*, *Salmonella* and *Enterobacter*, by modulating expression of AcrAB-TolC pump. In this study, the combined efflux-dependent resistance phenotype for the four antibiotics was found only in *K. pneumoniae* and *E. cloacae* complex, but not in *E. coli*. Among these 2 groups, analysis of the sequences of the genes encoding all the regulatory proteins involved in efflux pump regulation, as well as the structural genes of the pump proteins, shows that most of the mutations found in MDR strains are identified at the level of ramR gene and the ramAR intergenic region. This is not a new finding in these bacteria, as previous studies have shown that RamAR plays a major role in the overexpression of AcrAB-TolC efflux pumps, and mutations in RamR have also been identified. However, none of the previous studies analyzed such a large number of isolates and identified so many different mutations at ramAR locus. This work reinforces the observations of a number of previous publications, and has the advantage of having performed a sequence analysis of recently isolated clinical strains, listing mutations and deletions in the RamAR system that had not previously been described. The wide diversity of mutations identified suggests that clinical strains do not have a specific efflux-regulating response to antibiotics. Several evolutionary strategies are associated with RamR in MDR strains to reduce its repressive action, making RamA a key efflux regulator in the *Klebsiella-Enterobacter-Salomonella* group.

Specific remarks :

Introduction section:

-Line 77: ramAR has also been studied in *K. aerogenes* (Chollet et al. Antimicrob. Agents Chemother. 2004). Please note that ramAR does not exist in *E. coli* (mentioned delayed in the discussion).

Results section :

-Line 143 and legend of tables : Phe-Arg- β -naphthylamide (pABN) to be replaced by PABN

-Line 194: Interestingly, this deletion 1_69 was already observed in *E. cloacae* in a recent study (Ferrand A. et al., J Antimicrob Chemother. 2023)

Discussion section:

- Line 250: The genes of activators such as MarA, SoxS, RamA, etc. are not mutated, as these global regulators regulate various physiological processes and their modification would have a deleterious impact on the bacterium. Nevertheless, mutations have been regularly described in the repressors MarR (especially in *E. coli*), SoxR in various species and Rob.

- Line 277 : amino acid key position 154-155 was also of concern in *K. aerogenes* (Ref 19) and associated to resistance to chloramphenicol.

Reviewer #2 (Comments for the Author):

Gravey, Michel et al. examined ESBL-producing isolates collected from intensive care units for the so-called t2c2 phenotype (resistance to temocillin, tigecycline, ciprofloxacin and chlo-ramphenicol), which is due to mutations within the ramAR locus. The t2c2 phenotype was assessed using standard MIC assays and genome comparisons were based on Illumina se-quencing datasets.

The authors observed the t2c2 phenotype in 7% of all ESBL-E isolates analyzed, including 16 *K. pneumoniae* strains and 15 *Enterobacter cloacae* complex strains. Due to the reversibility of the t2c2 phenotype by the efflux pump inhibitor Phe-Arg- β -naphthylamide, the authors hy-potheseize a dominant role of RND pumps in the resistance pattern. In fact, the authors found various mutations in ramR, the intergenic ramAR region and acrR, which probably lead to a reduced suppression of acrAB expression and thus to an increased expression of the efflux pump.

The correlation shown by the authors between the t2c2 phenotype and the observed resistance patterns is interesting. Nevertheless, additional experiments should be used to support the authors' core statements. The comments can be found in the review document.

Specific Comments concerning the manuscript intitled : « Central rôle of the RamAR locus in the multidrug resistance in ESBL Enterobacterales ». Spectrum03548-23

General comments :

The authors screened 443 ESBL Enterobacteria for efflux resistance phenotype to 4 antibiotics (temocillin, tigecycline, ciprofloxacin, chloramphenicol), and identified those with mutations in the *ramAR* locus. The RamA regulatory protein is known to be primarily involved in regulating the active efflux of antibiotics in *Klebsiella*, *Salmonella* and *Enterobacter*, by modulating expression of AcrAB-TolC pump. In this study, the combined efflux-dependent resistance phenotype for the four antibiotics was found only in *K. pneumoniae* and *E. cloacae* complex, but not in *E. coli*. Among these 2 groups, analysis of the sequences of the genes encoding all the regulatory proteins involved in efflux pump regulation, as well as the structural genes of the pump proteins, shows that most of the mutations found in MDR strains are identified at the level of *ramR* gene and the *ramAR* intergenic region. This is not a new finding in these bacteria, as previous studies have shown that RamAR plays a major role in the overexpression of AcrAB-TolC efflux pumps, and mutations in RamR have also been identified. However, none of the previous studies analyzed such a large number of isolates and identified so many different mutations at *ramAR* locus.

This work reinforces the observations of a number of previous publications, and has the advantage of having performed a sequence analysis of recently isolated clinical strains, listing mutations and deletions in the RamAR system that had not previously been described. The wide diversity of mutations identified suggests that clinical strains do not have a specific efflux-regulating response to antibiotics. Several evolutionary strategies are associated with RamR in MDR strains to reduce its repressive action, making RamA a key efflux regulator in the *Klebsiella-Enterobacter-Salomonella* group.

Specific remarks :

Introduction section:

-Line 77: *ramAR* has also been studied in *K. aerogenes* (Chollet et al. Antimicrob. Agents Chemother. 2004). Please note that *ramAR* does not exist in *E. coli* (mentioned delayed in the discussion).

Results section :

-Line 143 and legend of tables : Phe-Arg- β -naphthylamide (pA β N) to be replaced by PA β N

-Line 194: Interestingly, this deletion 1_69 was already observed in *E. cloacae* in a recent study (Ferrand A. et al., J Antimicrob Chemother. 2023)

Discussion section:

- Line 250: The genes of activators such as MarA, SoxS, RamA, etc. are not mutated, as these global regulators regulate various physiological processes and their modification would have a deleterious impact on the bacterium. Nevertheless, mutations have been regularly described in the repressors MarR (especially in *E. coli*), SoxR in various species and Rob.

- Line 277 : amino acid key position 154-155 was also of concern in *K. aerogenes* (Ref 19) and associated to resistance to chloramphenicol.

Central role of the *ramAR* locus in the multidrug resistance in ESBL Enterobacterales

Gravey, Michel *et al.* examined ESBL-producing isolates collected from intensive care units for the so-called t2c2 phenotype (resistance to temocillin, tigecycline, ciprofloxacin and chloramphenicol), which is due to mutations within the *ramAR* locus. The t2c2 phenotype was assessed using standard MIC assays and genome comparisons were based on Illumina sequencing datasets.

The authors observed the t2c2 phenotype in 7% of all ESBL-E isolates analyzed, including 16 *K. pneumoniae* strains and 15 *Enterobacter cloacae complex* strains. Due to the reversibility of the t2c2 phenotype by the efflux pump inhibitor Phe-Arg- β -naphthylamide, the authors hypothesize a dominant role of RND pumps in the resistance pattern. In fact, the authors found various mutations in *ramR*, the intergenic *ramAR* region and *acrR*, which probably lead to a reduced suppression of *acrAB* expression and thus to an increased expression of the efflux pump.

Major comments

Lines 132-134: The authors mention that 16 *K. pneumoniae* strains and 15 *Ecc* strains had the t2c2 phenotype. These strains and their MIC values for TIG, CIP and CHL are also listed in Table 1. However, in the supplementary material (Table 2 and Table 3), the MIC test results are only listed for 11 ECC strains and 9 KP strains. Furthermore, the numbers are not consistent. If one compares the values for e.g. Kp20190611 in Table 1 and Table S2, it can be seen that the values for the inhibitor and the manual measurement are missing in the S2 table and that the MIC for CIP is 8 instead of 64 and for TIG 1 instead of 2.

Lines 143-147: The authors determined the MIC values manually and automatically and found differences in the size of 1 or 2 dilution steps. The treatment with the inhibitor is expressed in the text as "8-fold" etc., but it would be clearer for the reader to express it in dilution steps. A 2-fold reduction means a difference in one dilution step, which could be interpreted as normal variation. In addition, inhibitor treatment lowered the MICs, but some of the MIC values are still high and would be classified as resistant. Did the authors try different inhibitor concentrations? If not, this might be worth a try.

Lines 147-149: The authors should also repeat inhibitor tests for TEM. They argue that TEM MIC values are not affected by the inhibitor. However, this would be a great control to show that the inhibitor has no negative side effects on bacterial physiology, etc.

Lines 168-227: The authors explain in detail the mutations observed in the gene regions of interest, but there is no evidence that *acrAB* expression and thus the number of efflux pumps is increased in the selected isolates. In other publications in which the authors were involved (<https://doi.org/10.1128/aac.00358-23>), qPCR experiments were carried out to detect effects on expression, which would also be very helpful for this publication.

Lines 236-243: The authors point out the risk that the t2c2 phenotype is not recognized in diagnostic laboratories. However, CHL and TIG may not be the first-line agents for the treatment of Enterobacterales infections in Europe. Therefore, the significance of t2c2 detection may not be clear to the reader and should be explained in more detail.

Minor comments

An improvement of the language in general is recommended. The high number of spelling mistakes must be corrected.

Lines 155-156: Please reword, it should be made clear that this is not surprising information.

Line 190: Please avoid the wording "behavior of the protein".

Line 574: The authors should improve the resolution of their figures. The labeling of the axes must be larger.

Line 587, 597: The authors should improve the resolution of their figures.

Reviewer 1:

Remark 1: *Line 77: ramAR has also been studied in K. aerogenes (Chollet et al. Antimicrob. Agents Chemother. 2004). Please note that ramAR does not exist in E. coli (mentioned delayed in the discussion)*

Thank you very much for this precision, *Klebsiella aerogenes* has been added among the list of the species described in introduction with the appropriate reference. The fact that the locus *ramAR* locus does not exist among *Escherichia coli* is now clarified. The new version of the manuscript is : Line 74-77: “*ramAR* locus, which doesn’t exist in *Escherichia coli*, has been described widely among several species like *Salmonella enterica* serovar Typhimurium (7), *Enterobacter cloacae* complex (Ecc) (4), *Klebsiella pneumoniae* (8), and *Klebsiella aerogenes* whereas *marRAB* has been well studied in *E. coli* (9)”.

Remark 2: *Line 143 and legend of tables: Phe-Arg-β-naphthylamide (pAβN) to be replaced by PAβN.*

Thank you very much for this remark, the abbreviation pAβN have been correctly replace by PAβN as requested. Line 145 of the revised manuscript: “The addition of 20 mg/L of Phe-Arg-β-naphthylamide (PAβN) to the medium significantly decreased MICs of CHL, CIP and TIG (Table 1)”. Changed were also performed along the manuscript as well as in the Table I title and within the abbreviation’s descriptions.

Remark 3: *Line 194: Interestingly, this deletion I_69 was already observed in E. cloacae in a recent study (Ferrand A. et al., J Antimicrob Chemother. 2023).*

Thank you very much for this information. The appropriate reference has been added to the article and allow the support of the sentence in the discussion Lines 265 – 267 “First, the DNA-binding site could be altered by (i) massive deletions, as it has been found in *Ecc* strains in other studies (4, 20, 21), and (ii) amino acid substitutions ».

Remark 4: *Line 250: The genes of activators such as MarA, SoxS, RamA, etc. are not mutated, as these global regulators regulate various physiological processes and their modification would have a deleterious impact on the bacterium. Nevertheless, mutations have been regularly described in the repressors MarR (especially in E. coli), SoxR in various species and Rob.*

Thank you very much for this precision. The discussion part has been rewrite has followed: Lines 247-257: “In both species/complexes, sequences of CsrA, MarA, SoxS, SoxR, RamA, and MarR were conserved, whereas AcrR, *ramAR* IR and RamR showed the highest proportion of mutations. Consequently, *csrA*, *marA*, *soxS*, *soxR*, *ramA*, and *marR* could be considered as “stable” genes, probably because of their impact on several crucial cellular processes Indeed, CsrA plays a major role in intracellular carbon metabolism by controlling several essential enzymes involved in glucose metabolism in *E. coli* (17). *marRAB* and *soxRS* operons are both involved in oxidative stress responses (10, 18). It has been demonstrated that for clinical *E. coli* strains, the selection pressure constrains the mutations within the *marR* gene due to important fitness cost (19). Finally, mutations within the *soxRS* operon have been described among *E. coli* and *K. pneumoniae* strains ; they lead to cross antimicrobial resistance phenotype (20, 21) ».

Three new references, associated to this paragraph, have been added in the revised manuscript:

1. Praski Alzrigat, L., Huseby, D. L., Brandis, G., & Hughes, D. (2017). Fitness cost constrains the spectrum of *marR* mutations in ciprofloxacin-resistant *Escherichia coli*. *Journal of Antimicrobial Chemotherapy*, 72(11), 3016-3024.
2. Koutsolioutsou, A., Pena-Llopis, S., & Demple, B. (2005). Constitutive *soxR* mutations contribute to multiple-antibiotic resistance in clinical *Escherichia coli* isolates. *Antimicrobial agents and chemotherapy*, 49(7), 2746-2752.
3. Bialek-Davenet, S., Marcon, E., Leflon-Guibout, V., Lavigne, J. P., Bert, F., Moreau, R., & Nicolas-Chanoine, M. H. (2011). *In vitro* selection of *ramR* and *soxR* mutants overexpressing efflux systems by fluoroquinolones as well as cefoxitin in *Klebsiella pneumoniae*. *Antimicrobial agents and chemotherapy*, 55(6), 2795-2802.

Remark 5: Line 277: amino acid key position 154-155 was also of concern in *K. aerogenes* (Ref 19) and associated to resistance to chloramphenicol

Thank you very much for your report which supports our data. In the revised manuscript it is specify Lines 181 – 186 “These last have been described as a key positions for protein–substrate interactions in *Salmonella enterica* serovar Typhimurium (12). Amino acid replacement inside the 155 position or near of it (154) could change the behaviour of the protein, which will result in a lower functionality. Furthermore, deletions within the positions 154 and 155 of the RamR of a *K. aerogenes* strain was associated with high resistance to chloramphenicol (23) ».

Reviewer 2:

Remark 1: Lines 132-134: *The authors mention that 16 K. pneumoniae strains and 15 Ecc strains had the t2c2 phenotype. These strains and their MIC values for TIG, CIP and CHL are also listed in Table 1. However, in the supplementary material (Table 2 and Table 3), the MIC test results are only listed for 11 ECC strains and 9 KP strains. Furthermore, the numbers are not consistent. If one compares the values for e.g. Kp20190611 in Table 1 and Table S2, it can be seen that the values for the inhibitor and the manual measurement are missing in the S2 table and that the MIC for CIP is 8 instead of 64 and for TIG 1 instead of 2.*

Thank you very much your remark and taking the time to explore supplementary tables. Indeed, results of manual MICs were missing into the supplementary tables two and three. All the data have been carefully added into the appropriate columns into supplementary tables two and three. This process allows to correct one mistake which was into the table I for the MIC of chloramphenicol for the strain Kp20191103. Importantly, MICs presented into the Table I are those obtained manually. According to the limited ranges of the MICs from the sensititre, it could be a variation between MICs according to the method used.

Remark 2: Lines 143-147: *The authors determined the MIC values manually and automatically and found differences in the size of 1 or 2 dilution steps. The treatment with the inhibitor is expressed in the text as "8-fold" etc., but it would be clearer for the reader to express it in dilution steps. A 2-fold reduction means a difference in one dilution step, which could be interpreted as normal variation. In addition, inhibitor treatment lowered the MICs, but some of the MIC values are still high and would be classified as resistant. Did the authors try different inhibitor concentrations? If not, this might be worth a try.*

Thank you very much for your remarks. Replacement between fold change and dilutions have been made in the text. In the revised manuscript, results are now presented has follow: lines 146-148 “CHL showed the greatest MIC reduction, between 3- and 4-dilutions, followed by CIP (between 0- and 4-dilutions) and TIG (between 1- and 3-dilution) (Table 1).”

It is true that inhibitor treatment lowered the MICs and sometimes drugs were still classified as resistant. In our opinion, the limitation effect of PAβN is the result of complex antimicrobial resistance phenotypes. Indeed, excepted the Ecc20200803 strain, all strains had additional fluoroquinolones resistance mechanisms of which the PAβN is inefficient. Regarding chloramphenicol, excepted the strains which add *cat* of *floR* genes, all the strains were classified

sensible to this antibiotic with 20 mg/L of PA β N. It is true that, for tigecycline, some measured MICs with 20 mg/L of PA β N were equal to 1 mg/L, nevertheless, in all the studied strains, there was a MICs decreased between one and three dilutions. For the strains with a remaining MICs at 1 mg/L, it could be hypothesised that, RND pump dysregulation was not the only mechanism involved in the resistance observed.

Only one concentration of PA β N was used during this study. The concentrations 20/25 mg/L were also used in recent studies

- Guérin, F., Lallement, C., Isnard, C., Dhalluin, A., Cattoir, V., & Giard, J. C. (2016). Landscape of resistance-nodulation-cell division (RND)-type efflux pumps in *Enterobacter cloacae* complex. *Antimicrobial agents and chemotherapy*, 60(4), 2373-2382;
- Vera-Leiva, A., Carrasco-Anabalón, S., Lima, C. A., Villagra, N., Domínguez, M., Bello-Toledo, H., & González-Rocha, G. (2018). The efflux pump inhibitor phenylalanine-arginine β -naphthylamide (PA β N) increases resistance to carbapenems in Chilean clinical isolates of KPC-producing *Klebsiella pneumoniae*. *Journal of Global Antimicrobial Resistance*, 12, 73-76,
- Ferrand, A., Vergalli, J., Bosi, C., Pantel, A., Pagès, J. M., & Davin-Regli, A. (2023). Contribution of efflux and mutations in fluoroquinolone susceptibility in MDR enterobacterial isolates: a quantitative and molecular study. *Journal of Antimicrobial Chemotherapy*, 78(6), 1532-1542.

It is true that more important concentrations could be interesting regarding tigecycline resistance, but we didn't try.

Remark 3: *Lines 147-149: The authors should also repeat inhibitor tests for TEM. They argue that TEM MIC values are not affected by the inhibitor. However, this would be a great control to show that the inhibitor has no negative side effects on bacterial physiology, etc.*

Thank you very much. As mentioned above, PA β N concentrations chosen have previously been used in several studies without any consequences regarding bacterial physiology. About the TEM MICs in presence of PA β N, interpretation of β -lactams MICs in the presence of PA β N could be difficult as the PA β N has an effect of porins expression OmpC and OmpF. As the consequence an elevation of β -lactams MICs can be observed with PA β N:

- for temocillin as mentioned in our previous article “*ramR* deletion in an *Enterobacter hormaechei* isolate as a consequence of therapeutic failure of key antibiotics in a long-term hospitalized patient. *Antimicrobial agents and chemotherapy*, 64(10), 10-1128”

- or for carbapenems which was reported in this article “The efflux pump inhibitor phenylalanine-arginine β -naphthylamide (PA β N) increases resistance to carbapenems in Chilean clinical isolates of KPC-producing *Klebsiella pneumoniae*. *Journal of Global Antimicrobial Resistance*, 12, 73-76.”

Remark 4: *Lines 168-227: The authors explain in detail the mutations observed in the gene regions of interest, but there is no evidence that *acrAB* expression and thus the number of efflux pumps is increased in the selected isolates. In other publications in which the authors were involved (<https://doi.org/10.1128/aac.00358-23>), qPCR experiments were carried out to detect effects on expression, which would also be very helpful for this publication.*

Thank you very much, you were completely right. To eliminate this limitation, RNA extractions and qRT-PCR were performed on each mutation found within the *ramAR* intergenic region and in the *RamR* sequence. Results are presented in the Table 3 and the results section has been change regarding the new results.

Remark 5: *Lines 236-243: The authors point out the risk that the *t2c2* phenotype is not recognized in diagnostic laboratories. However, *CHL* and *TIG* may not be the first-line agents for the treatment of *Enterobacterales* infections in Europe. Therefore, the significance of *t2c2* detection may not be clear to the reader and should be explained in more detail.*

We are right with the reviewer’s comment. It is not necessary to highlight the under-estimation of the *t2c2* phenotype due to different panel and/or categorization of antimicrobial susceptibility testing among countries. As it represents a minor comment, we prefer to delete this point in our Manuscript

Minor comments:

Remark 6: *An improvement of the language in general is recommended. The high number of spelling mistakes must be corrected.*

We are sorry about the presence of spelling mistakes; the manuscript has been carefully corrected.

Remark 7: *Line 155—156 Please reword, it should be made clear that this is not surprising information.*

Thank you very much, the sentence has been rewritten, and is now: Line 157-158 “. As expected, the *ramAR* operon was absent from the 194 genomes of *E. coli* studied.”

Remark 8: *Line 190: Please avoid the wording "behavior of the protein".*

Thank you very much, the expression “behaviour of protein” has been removed from the revised manuscript.

Line 574: The authors should improve the resolution of their figures. The labelling of the axes must be larger. & Line 587, 597: The authors should improve the resolution of their figures.

Thank you very much for these remarks, labelling of the axes has been improved for the figure 1 and resolution of all the figures was increased according to the author’s guidelines.

Re: Spectrum03548-23R1 (Central role of the *ramAR* locus in the multidrug resistance in ESBL *Enterobacteriales*)

Dear Dr. Francois Gravey:

Thank you for revising your manuscript. I have received reviews (attached) and another round of revisions is necessary before your manuscript can be reconsidered for publication. While Reviewer 1 was satisfied with the changes, Reviewer 2 has indicated the need for additional editing and has expressed concerns about the presentation of the manuscript. Please ensure that you proofread the manuscript carefully before submitting it again and address all of the comments noted below.

Revision Guidelines

Sincerely,
Shannon Manning
Editor
Microbiology Spectrum

Reviewer #2 (Comments for the Author):

Dear authors,

Thank you for processing my notes and comments. In my opinion, the language and the resolution of an illustration are still not good enough and should be revised again. I would like to provide you with the new notes below as a rough guide.

Line 71: „associated with" was correct, please change it again.

Line 81: I think "organized in" would be correct.

Line 125: Better: "These data show a great diversity of bacterial populations and resistomes among the strains analyzed."

Line 138: resistant

Line 150: dilutions

Lines 157-160: better: The sequences of twelve genes linked to the expression of RND pumps (ramA, ramR, acrR, acrA, acrB, tolC, soxR, soxS, marA, marB, marR, and csrA and the ramAR intergenic region (IR)) were found among the 122 K. pneumoniae and 127 Ecc strains.

Line 169: within a regulatory pathway

Line 186: fold change

Line 245: massive

Line 249-251: better: It is noteworthy that some genomic alterations led to a stronger dysregulation of ramA, acrA and tolC than others, suggesting an unequal consequence on the integrity and functionality of RamR.

Line 449-452: Better: The expression ratios of ramA, acrA and tolC in strains with a t2c2 phenotype were determined by comparison with the transcription levels of the K. pneumoniae wildtype strain XXX.

Please mention which kind of WT strain you have used!

Line 632 (and in general): Please write out numbers up to twelve as a word if they are not followed by a unit.

Line 630: Figure 1 is still too blurred; you will need to create the image in a different program or increase the resolution when exporting.

Line 654-655: Please change this part: "no significative shit".

Line 656-657: Correct the sentence.

Line 683: Please provide standard deviations.

Reviewer 2:

Line 71: *« associated with » was correct, please change it again.*

Thank you very much, it has been changed again.

Line 81: *I think "organized in" would be correct.*

Thank you very much, the correct expression has been added in the text.

Line 138: *resistant*, Line 150: *dilutions*, Line 186: *fold change*, Line 245: *massive*

Thank you, please excuse these typing errors.

Line 125: Better: *"These data show a great diversity of bacterial populations and resistomes among the strains analyzed."*

Lines 157-160: better: *« The sequences of twelve genes linked to the expression of RND pumps (ramA, ramR, acrR, acrA, acrB, tolC, soxR, soxS, marA, marB, marR, and csrA and the ramAR intergenic region (IR)) were found among the 122 K. 160 pneumoniae and 127 Ecc strains. »*

Line 249-251: better: *« It is noteworthy that some genomic alterations led to a stronger dysregulation of ramA, acrA and tolC than others, suggesting an unequal consequence on the integrity and functionality of RamR ».*

Thank you very much, all the suggested sentences are now included in the manuscript.

Line 169: *“ within a regulatory pathway ”*

Thank you, the expression has been added.

Line 449-452: Better: *“The expression ratios of ramA, acrA and tolC in strains with a t2c2 phenotype were determined by comparison with the transcription levels of the K. pneumoniae wildtype strain XXX”.* Please mention which kind of WT strain you have used!

Thank you very much for the suggestion, the sentence as been changed considering your remarks. The *Klebsiella pneumoniae* strain is now completely described in the method section, subsection “RNA extraction and RT-qPCR” Lines 433 – 437 “The expression ratios of *ramA*, *acrA* and *tolC* in strains with a t2c2 phenotype were determined by comparison with the transcription levels of the *K. pneumoniae* strain from our lab P013-01-S1 which had a wild-type antimicrobial resistance phenotype profile. Genome of the strain ant its antibiogram are available under the following accession number: SAMN41383518. »

Line 632 (and in general): Please write out numbers up to twelve as a word if they are not followed by a unit.

Thank you, all the numbers up to twelve which were not included in a numeric expression have been written in words.

Line 630: Figure 1 is still too blurred; you will need to create the image in a different program or increase the resolution when exporting.

The figure 1 was exported with another software.

Figure legends:

Line 654-655: Please change this part: "no significative shit".

Line 656-657: Correct the sentence.

The legends of the figure have been corrected for the figure 3 and 4.

Table III

Line 683: Please provide standard deviations.

The standard deviations have been added in the table III.

Re: Spectrum03548-23R2 (Central role of the *ramAR* locus in the multidrug resistance in ESBL *Enterobacterales*)

Dear Dr. Francois Gravey:

Thank you for submitting another revision. The manuscript is much improved over the original version and hence, I am recommending it for acceptance. My only concern is that the sequencing data have not been released. Please understand that this is necessary prior to publication (see below for more information). While I appreciate you providing the Bioproject and sequencing accession numbers, a search of these numbers did not return the full records.

At this time, I am forwarding it to the ASM production staff for publication. Your paper will first be checked to make sure all elements meet the technical requirements. ASM staff will contact you if anything needs to be revised before copyediting and production can begin. Otherwise, you will be notified when your proofs are ready to be viewed.

Sincerely,
Shannon Manning
Editor
Microbiology Spectrum